# Effects of a 10-Week Physical Activity Intervention on Asylum Seekers’ Physiological Health

**DOI:** 10.3390/brainsci12070822

**Published:** 2022-06-24

**Authors:** Matheus Guerra, Danilo Garcia, Maryam Kazemitabar, Erik Lindskär, Erica Schütz, Daniel Berglind

**Affiliations:** 1Department of Global Health, Karolinska Institute, 171 77 Stockholm, Sweden; daniel.berglind@ki.se; 2Promotion of Health and Innovation (PHI) Lab, International Network for Well-Being, New Haven, CT 06510, USA; maryam.kazemitabar@yale.edu (M.K.); erik.lindskar@regionblekinge.se (E.L.); erica.schutz@lnu.se (E.S.); 3Department of Behavioral Sciences and Learning, Linköping University, 581 83 Linköping, Sweden; 4Centre for Ethics, Law and Mental Health (CELAM), University of Gothenburg, 405 30 Gothenburg, Sweden; 5Department of Psychology, Lund University, 221 00 Lund, Sweden; 6Department of Psychology, University of Gothenburg, 405 30 Gothenburg, Sweden; 7School of Public Health, Yale University, New Haven, CT 06510, USA; 8Department of Psychology, Linnaeus University, 450 85 Kalmar, Sweden; 9Center for Epidemiology and Community Medicine (CES), Region Stockholm, 104 31 Stockholm, Sweden

**Keywords:** physical activity, intervention, asylum seekers, physiological health, VO_2_ max

## Abstract

Introduction: The rise in armed conflicts has contributed to an increase in the number of asylum seekers. Prolonged asylum processes may negatively affect asylum seekers’ health and lead to inactivity. Studies show that physical activity interventions are associated with improvements in health outcomes. However, there are a limited number of studies investigating the associations of physical activity on asylum seekers’ health. Methods: Participants (263 males and 204 females), mostly from Syria, were assessed before and after a 10-week intervention for VO_2_ max, body mass index (BMI), skeletal muscle mass (SMM), body fat, and visceral fat. Linear mixed models were used to test differences within groups, and a linear regression model analysis was performed to test whether physiological variables predicted adherence. Results: Participants’ VO_2_ max increased: males by 2.96 mL/min/kg and females 2.57 mL/min/kg. Increased SMM percentages were seen in both genders: females by 0.38% and males 0.23%. Visceral fat area decreased: males by 0.73 cm^2^ and females 5.44 cm^2^. Conclusions: Participants showed significant increases in VO2 max and SMM and decreased visceral fat. This study provides an insight into asylum seekers’ health and serves as a starting point to new interventions in which physical activity is used as a tool to promote and improve vulnerable populations’ health.

## 1. Introduction

One of the biggest challenges in the 21st century is the increasing number of armed conflicts in the world [1], which has contributed to an increased number of displaced populations and asylum seekers. In 2015, about 163,000 individuals sought asylum in Sweden, compared to an average of 28,575 per year during 2000–2010 [2]. In Blekinge, Sweden, the number of asylum seekers registered in the county’s five municipalities in 2015 amounted to a total of 4069—which represents a significant increase when compared to the previous years (2014: 2126; 2013: 1280; and 2012: 1032) [2]. Lengthy application processes, which in some cases can take several years to complete, may adversely affect the physical and mental health of asylum seekers [3]. Additionally, other difficult and unusual situations, combined with limited language skills, may prevent asylum seekers from receiving the information required to navigate their newly adopted society [4].

Even though asylum seekers are a very diverse group regarding health status, research shows that they generally have more health problems, such as lower physical and mental health, than native populations [5,6,7]. For example, a previous study with an Iraq-born population in Sweden found a higher level of physical inactivity and a higher risk of the development of non-communicable diseases such as diabetes type 2 when compared with native Swedes [8]. Other studies have also found that Middle Eastern immigrants in Sweden have a four-fold higher risk of developing diabetes when compared with native Swedes [9], which can be partly attributed to a high prevalence of obesity in non-European immigrants [10]. Consequentially, there is a need to increase and promote physical activity among foreign-born populations in Sweden.

It is a well-known fact that low cardiorespiratory fitness is one of the leading risk factors causing non-communicable diseases and overall mortality, while physical activity contributes to physical and mental well-being and increases the possibilities for creating social networks as well as being part of society [11]. For example, higher cardiorespiratory fitness is prospectively associated with lower all-cause mortality, in which a 1.0 mL/min/kg higher maximal oxygen consumption (VO_2_ max) is associated with a 9% relative risk reduction in all-cause mortality [12].

Moreover, the loss of skeletal muscle mass, together with the excessive gain of fat mass, is associated with several metabolic disorders such as metabolic syndrome, diabetes, and cardiovascular diseases [13]. However, skeletal muscle loss is, to a large extent, reversible through the adoption of resistance training and diet [14]. According to research conducted by Roth et al. [15], physical activity and resistance training are effective for the prevention of the loss of skeletal muscle mass which, in turn, improves quality of life.

Last but not the least, excess visceral fat is closely associated with the development of many non-communicable diseases such as hypertension, type 2 diabetes, and hyperlipidemia and constitutes an independent risk factor for developing heart disease [16]. Diet and regular exercise have been shown to be effective in reducing visceral fat, and the association between physical exercise and the reduction in visceral fat volume has been well established [17,18].

Despite the vast amount of evidence linking physical activity to improved quality of life and a reduced outcome of non-communicable diseases [19,20,21], there are few studies exploring the associations of physical activity interventions with asylums seekers’ health [22,23]. One such study [24] investigated the impact of an eight-week training program in 45 young males (mean age = 25.6, SD = 7.1) living in a refugee camp in Greece. The participants were invited to engage in physical activities three to five times per week for approximately one hour, focusing on a combination of weight and endurance training. The study found that higher participation rates were associated with fewer anxiety symptoms, higher health-related quality of life, higher self-perceived fitness, greater handgrip strength, and improved cardiovascular fitness.

The project “Health for Everyone-Sport, Culture and Integration” was an initiative created by the Ronneby municipality in Blekinge, Sweden, in partnership with the Blekinge Sports Association. Within the project, asylum seekers were invited to engage in physical activity once a week during a 10-week period in groups of 20 to 30 individuals. In addition to this, participants were invited to a once-a-week class on health promotion in their native language and a visit to the Blekinge Museum in order to introduce them to Scandinavian history and Blekinge’s cultural heritage.

The aim of this study was to evaluate whether there were any significant changes in physiological health among asylum seekers who participated in the “Health for Everyone” project and to investigate whether physiological health measurements at baseline predicted adherence to the intervention. We expected to see improvements in most of the physiological health measures and hypothesized that individuals who were more physically fit at baseline would have higher rates of attendance.

## 2. Materials and Methods

### 2.1. Participants and Procedure

The project “Health for Everyone-Sport, Culture and Integration” was created by the Ronneby municipality and carried out in partnership with the Blekinge Sports Association. It started in the fall of 2016 and lasted for two years, with a total of 18 months of active operation. The recruitment was carried by Blekinge’s municipalities, being scheduled within the framework of the social orientation for asylum seekers from countries currently experiencing armed conflicts. In total, 467 individuals (263 males and 204 females) with a mean age of 35.9 years (SD = 11.9) were enrolled in the project.

The participants engaged in a combination of resistance and aerobic training designed in a circuit format alternating different exercise stations. The participants were provided with transportation from their settlements in each of the municipalities to the training facility in Karlskrona (Blekinge Health Arena) for the once-a-week training session.

At the beginning of the intervention, the participants received information (verbally and written) about the activities in Arabic, Somali, and Persian languages, with the assistance of community-appointed translators, and they were subsequently asked to participate in the evaluation carried out by the research group. The participants were informed that their data were confidential, and that the data would be used for scientific analysis and publication. All the participants in this study gave their consent to participate in writing.

The participants underwent physiological tests consisting of a bioelectrical impedance measurement using an InBody 720 body composition analyzer (Biospace Co., Ltd., Seoul, Korea) to measure body weight, body mass index (BMI), skeletal muscle mass (SMM), body fat, visceral fat, and cardiorespiratory fitness (VO_2_ max) through a beep test both at the beginning of the study and at the end of the intervention. The participants were also asked to answer questions regarding their background (demographical data, age, and gender, etc.) and other self-reports of validated psychological measures. In total, the data collection took approximately 1.5 h and was performed at baseline (week 0) and at endpoint (week 10), leading to a total of 8 training sessions within the intervention.

### 2.2. Measures

#### 2.2.1. Attendance

Attendance was logged by the Blekinge Health Arena instructors on each scheduled once-a-week training day. A total attendance of eight times was the maximum attendance rate. Therefore, we divided the number of recorded attendances for each participant by eight, which gave us the attendance percentage for each participant in the project.

#### 2.2.2. Cardiorespiratory Fitness

The multi-stage fitness test, also known as the beep test, was used to estimate the participants’ cardiorespiratory fitness (i.e., VO_2_ max). The test has been widely used due to its simplicity in providing an accurate approximation of an individuals’ VO_2_ max [25]. The test requires participants to run 20 m back and forth across a marked track, keeping time with beeps. Every minute, the next level starts; the time between beeps gets shorter, which requires participants to run faster to keep up with the next level. If the participant fails to reach the relevant marker in time, a first warning will be given, with a second warning meaning the end of the test. The number of shuttles successfully completed is therefore registered, and the final score is given according to which level and the total number of shuttles the participant was able to complete. The following formula is used to transform the beep test results to VO_2_ max: VO_2_ max = 3.46 × (Level + No. of Shuttles/(Level × 0.4325 + 7.0048)) + 12.2.

#### 2.2.3. Body Weight and Body Mass Index (BMI)

BMI is a statistical index calculated by a person’s weight divided by height in square meters or BMI = weight (kg)/height^2^ (m) [26]. The number obtained by the equation is the individual’s BMI, and it is used to define an individual as underweight, normal weight, overweight, or obese. A higher BMI indicates a higher likelihood of obesity. A commonly used reference range for normal weight is between 18.5 and 24.9 kg/m^2^ [26].

The BMI calculation was performed using measures obtained from a direct segmental multi-frequency bioelectrical impedance analysis (DSM-BIA) with an InBody 720 body composition analyzer. The DSM-BIA technique provides an accurate assessment of segmental and body composition [27,28].

#### 2.2.4. Skeletal Muscle Mass (SMM)

In humans, skeletal muscle is a type of striated muscle tissue which is under voluntary control of the somatic nervous system. It constitutes approximately 40% of the total body mass [29] and can be influenced by a person’s nutritional status, hormonal balance, physical activity levels, or disease. The SMM calculation was conducted using measures obtained from a DSM-BIA with an InBody 720 body composition analyzer.

#### 2.2.5. Body Fat Mass

Body fat mass refers to the amount of adipose tissue that constitutes the human body. The excessive accumulation of fat represents obesity, typically classified through the BMI with the underlying assumption that a higher BMI indicates increased body fat. However, BMI does not measure body fat mass directly and provides no information on the location of fat mass in different body sites. Hence, as a complement, body fat mass calculation was conducted using measures obtained from a DSM-BIA with an InBody 720 body composition analyzer.

#### 2.2.6. Visceral Fat

Visceral or abdominal fat refers to adipose tissue accumulated in the abdominal cavity between internal organs such as the liver, stomach, and intestines. The cut-off value of visceral fat area associated with an increased risk of obesity-related disorder, according to the receiver operating characteristics curve, was 103.8 cm^2^ [30]. The visceral fat calculation was conducted using measures obtained from a DSM-BIA with an InBody 720 body composition analyzer.

### 2.3. Statistical Analysis

First, we removed the outliers. Outliers are values that deviate remarkably from other values [31], which make data distribution non-normal and create significant changes in parameter estimates, especially when the maximum likelihood estimation method is used [32]. In this study, outliers were detected using boxplots and scatterplots. A total of 93 extreme outliers in the variables of VO_2_ max, body weight, BMI, skeletal muscle mass (%), body fat mass (%), and visceral fat were removed in order to acquire normal distribution of the data. Then, the normality of the data was measured by investigating the skewness and kurtosis of the variables. All these values were within the range of ±1, and therefore, we considered the data distribution as normal for the dependent variables. Additionally, an MCAR test was conducted, and the results showed that the missing data were completely at random (Chi-Square = 51.65, df = 52, *p* = 0.49).

The study had an interventional and longitudinal design. In short, a set of physiological variables was measured before and after the 10-week physical activity intervention for each participant. We used two linear mixed models to test differences within groups with regard to physiological health variables (i.e., cardiorespiratory fitness, body weight, BMI, SMM%, body fat mass%, and visceral fat) at the start (T1) and end (T2) of the intervention. In Models 1 and 2, T1 and T2 measures of physiological health were entered as dependent variables. In Model 2, gender, age, and attendance percentage were included as covariates into the regression model. The covariates’ intercepts effects were fixed, and individuals’ intercepts were set at random to test the differences within individuals with regard to the dependent variables.

We used the restricted maximum likelihood estimation method, which provides more accurate and unbiased results compared to other methods [33]. In addition, we used intraclass correlation coefficients (ICCs) as a measure of the variance explained by individuals; that is, an estimation of the group mean reliability across T1 and T2 [34]. This was the ratio of between-group variance to total variance. It was calculated using the following formula for each linear regression model:$$ICC=\frac{\sigma_{0}^{2}}{\sigma_{0}^{2}+\sigma_{e}^{2}}$$
in which $\sigma_{0}^{2}$ is the variance of random intercept and $\sigma_{0}^{2}+\sigma_{e}^{2}$ is the total variance (i.e., random intercept variance and residual variance). The result is usually between 0 and 1; higher values suggest greater between-group variability.

As the last analysis, we performed a linear regression model analysis to test whether variables in physiological health at baseline (i.e., T1) could predict adherence to the physical intervention (i.e., attendance percentage). All the statistical analyses were conducted using IBM SPSS Statistics v.26 software, and statistical significance for all analyses was set at *p* < 0.05. The statistical power (1 − *β*) for the total sample at *α* = 0.05 was equal to 0.99.

## 3. Results

### 3.1. Sample Characteristics

Table 1 indicates the descriptive characteristics of the sample used in this study. The comparison of the mean differences in attendance percentage among females and males indicated that both genders participated in the physical activity sessions to roughly the same extent. Table 2 indicates the physiological health variables related to before (T1) and after (T2) the intervention.

### 3.2. Linear Mixed Model Analysis: Effect of the Intervention

Table 3 shows the results of Models 1 (null model), 2 (random effects model), and 3 (fixed effects model) regarding predictors of physiological health. For Model 1, the results indicated that there were differences in physiological health measures between T1 and T2 within individuals (*p* < 0.00) f or all the various physiological measures (i.e., cardiorespiratory fitness, body weight, BMI, SMM%, body fat mass%, and visceral fat). In Model 2 (random effects model), we estimated the differences within individuals regarding physiological health measures in T1 and T2 by controlling for gender, age, and attendance percentage as predictors in the equation and putting individuals as random effects. The results showed that the predictor variables in Model 2 (i.e., gender, age, and attendance percentage) significantly affected the intercepts of the dependent variables in Model 1 (i.e., the physiological health measures). The ICCs for all the equations showed that the added predictors in Model 2 changed the independent intercepts in Model 1.

To be more precise, the results of the linear mixed model for Model 2 showed that the changes from T1 to T2 regarding cardiorespiratory fitness, BMI, body weight, and SMM percentage were significant (*p* ≤ 0.01). Gender predicted changes in body fat percentage and SMM percentage (*p* < 0.05) from T1 to T2. Thus, this suggests that females had a greater relative increase in SMM percentage and a greater relative decrease in body fat percentage compared to males.

All the intercepts for visceral fat and the predictors were non-significant (*p* > 0.05). Hence, there were no differences within individuals concerning visceral fat, and gender, age, and attendance percentage did not predict changes in visceral fat values. Importantly, attendance percentage did not have any association with changes in physiological health variables.

For a comparison of the fixed versus random effects models, the fixed effects model was also measured. The results of the linear mixed model for Model 3 (fixed effects model) yielded similar outputs as the random effects model with several differences. In the fixed effects models, age predicted cardiorespiratory fitness, body fat percentage, and SMM percentage. In addition, gender predicted cardiorespiratory fitness in addition to those significant relationships in the random effects model. The comparison of the Akaike Information Criteria (AIC) showed that the random effects model better fit to the data. In this study, the outputs of the random effects model were considered for further analysis with regard to the model’s advantages over the fixed effects model. The random effects model is capable of estimating shrunken residuals [35] and provides estimates that overall are closer to the true value in any particular sample [36].

### 3.3. Effect Size and Minimum Detectable Change Calculation for Each Physiological Measure

Table 4 indicates the effect size Cohen’s *f^2^* for each measure across females and males, which were calculated using GPower v3.1. The level of effect sizes was small for VO_2_ max, skeletal muscle mass, body fat mass for females, and visceral fat. The effect sizes for body weight, BMI for both females and males, and body fat mass for males were not significant. Cohen [37] suggested cut-off points of *f*
^2^ ≥ 0.02, *f^2^* ≥ 0.15, and *f*
^2^  ≥ 0.35 representing small, medium, and large effect sizes, respectively. Moreover, small differences in effect sizes between females and males with respect to skeletal muscle mass and body fat mass were observed. Changes in physiological health measures were also estimated through minimum detectable change (MDC)—a statistical estimate of the smallest amount of change that can be detected by a measure that corresponds to a noticeable change in the variable under study over time which is not related to measurement error. MDC is calculated using the following formula:MDC = SEM × 1.96 × square root of 2
$$\mathrm{SEM}=\mathrm{SD}\;\times \;\sqrt[{2}]{(1-r})$$
where 1.96 is a *z*-score which represents the confidence interval from a normal distribution, SD is the standard deviation at baseline, *r* is the test–retest reliability coefficient, and SEM is the standard error of measurement.

The MDC value was assumed to be the minimum amount of change that needs to be observed so that it could be considered a real change [38] or a change to which the amount of change in performance was likely to be greater than the amount of random measurement error. Since the MDC_95_ values were greater than the SEM values, the changes in physiological health measures were not related to measurement error and therefore could be considered as real changes. Moreover, the visceral fat’s SEM was greater than MDC_95_, thus being consistent with the results from the linear mixed method in which the changes in visceral fat were not significant. Finally, although the linear mixed analysis showed that changes in body fat percentage were not significant, the MDC_95_ was greater than the SEM values for this variable. Perhaps this reflects the significant body fat percentage reduction in females shown using the linear mixed model method.

### 3.4. Regression Analysis: Baseline Physiological Health as Predictor of Attendance

We conducted a linear regression analysis to investigate whether physiological health variables at baseline (T1) as well as age and gender predicted the attendance percentage. As expected, cardiorespiratory fitness at baseline was significantly and positively (*F*_(1, 307)_ = 9.25, *p* < 0.01, adj. *R*^2^ = 0.03) related to asylum seekers’ attendance to the physical intervention sessions (*p* < 0.01). Nevertheless, this correlation was relatively low (*r* = 0.17, *p* < 0.01). See Table 5 for details.

## 4. Discussion

The study’s aim was to evaluate the significance of the 10-week physical activity intervention, “Health for Everyone”, on asylum seekers’ physiological health. The results of the linear mixed model confirmed that the “Health for Everyone” intervention was associated with a beneficial impact on the asylum seekers’ physiological variables in both males and females, with improvements in their body composition and cardiorespiratory fitness.

In general, the participants showed a significant increase in cardiorespiratory fitness, with females showing a decrease in their total body weight, while males showed a slight increase. Both genders showed an increase in skeletal muscle mass and a decrease in body fat mass percentages and in visceral fat area. For all the participants, the total attendance percentage for the duration of the program was 73%. Using minimum detectable changes (MDC) and effect sizes, we found significant differences in the physiological health variables over time. This justifies the changes we obtained from the linear mixed model analysis.

Similar results have been presented in a systematic review and meta-analysis of 65 physical activity interventions [39], showing significant improvements in the cardiometabolic health of the participants. From baseline to post-intervention, short-term interventions of <12 weeks significantly improved cardiorespiratory fitness in populations with overweight/obesity, along with a decrease in cardiometabolic risk factors.

Moreover, in a culturally adapted lifestyle intervention, including seven sessions addressing healthy diet and physical activity [40] in a sample of 96 Iraq-born immigrants residing in Malmö, Sweden, beneficial effects on insulin levels, body weight and LDL cholesterol were found.

### 4.1. Cardiorespiratory Fitness

The participants showed significant increases in cardiorespiratory fitness from T1 (week 0) to T2 (week 10). At the end of the intervention, males showed an increase in VO_2_ max of 2.90 mL/min/kg (M = 36.44, SD = 6.49) and females an increase of 2.64 mL/min/kg (M = 31.22, SD = 4.87). For comparison, a population-based study of 579 men aged 42 to 60 [12] found that a 1.0 mL/min/kg increase in VO_2_ max was prospectively associated with a 9% risk reduction in all-cause mortality, emphasizing the importance of increasing and maintaining cardiorespiratory fitness levels to promote long-term health.

Albeit an increase in cardiorespiratory fitness was seen, and considering the participants’ mean age of 35.9 (SD = 11.9), the results show that males in our study population had lower cardiorespiratory fitness levels when compared to females. Using the cardiorespiratory fitness classification scale for the age group of 30–39 years old [41], males were classified as having poor cardiorespiratory levels at T1 and T2, with females being classified as having fair cardiorespiratory fitness level at both points in time. It is worth noting that although both groups remained in the same cardiorespiratory fitness classification category, males and females increased their VO_2_ max during the 10-week intervention period.

Comparing our sample’s results with the Swedish population, a study [42] of relative VO2 max trends in the working force from 1995 to 2017 (*N* = 354 277 participants; 44% women, 56% men, aged 18–74 years) showed that within the age group of 35–49 years, females had a mean relative VO_2_ max of 35.7 mL/kg/min (SD = 1.23) and males a mean of 34.9 mL/kg/min (SD = 1.38). Hence, at least for the males in our asylum seeker sample, their VO_2_ max improved and reached similar levels to native Swedes.

### 4.2. Skeletal Muscle Mass

In our study, both genders increased their skeletal muscle mass percentages from baseline to endpoint. While female participants at endpoint showed an increase of 0.38%, male participants showed an increase in skeletal muscle mass percentage of 0.24% from T1 to T2. Contrasting with the European native population, a study [43] including a total of 1664 Hungarian adults (1198 females and 466 males) found that the mean skeletal muscle mass for the age group of 20–40 years was 46.51% and 39.60% for males and females, respectively. Another investigation with a sample of 1 924 Serbian women with a mean age of 35.5 years [44] registered the average skeletal muscle mass for the total sample as 39.3%.

Bearing in mind that our study’s population consisted of foreign-born asylum seekers, and although skeletal muscle mass percentages increased for both genders, our results showed considerably lower skeletal muscle mass percentages when compared to the native European population described above.

### 4.3. Body Fat

Despite the relatively limited number of training sessions offered to the participants, both genders showed a reduction in total body fat percentages, with males registering a decrease of 0.35% at T2 and females a decrease of 0.73%. At endpoint, females reduced more in body fat percentages than their male counterparts. Although both genders showed a reduction in percentual points, males and females still showed comparably higher percentages at T2 when compared to European populations. Ihász et al. [43] showed that the mean body fat percentages in the age group of 20–40 years were 18.27% for males and 27.75% for females. In a study [45] consisting of 433 healthy Caucasians (253 men and 180 women) aged 18–94 years, in the age group of 35–59 years, the fat mass percentages values in males were determined to be 21.2% and 29.0% for females. Moreover, the American College of Sports Medicine [41], in their classification scale for body composition, states for the age group of 30–39 years that *good* body fat percentages are 15.9–18.4% for men and 17.5–21.0% for women.

In this context, the males in our study presented poor body fat percentages followed by very poor percentages for females at T1 and T2, while the above-mentioned studies sampling European populations presented fair/good percentages for males and poor/fair percentages for females, according to their respective age groups.

### 4.4. Visceral Fat

The current literature maintains that healthy levels for visceral fat area should be sustained at <100 cm^2^, with values of ≥100 cm^2^ associated with an increased risk of obesity-related disorders such as hypertension, hyperglycemia, and dyslipidemia [46]. In a study of 413 subjects (174 men and 239 women) to determine cut-off values for visceral fat area associated with an increase in the risk of obesity disorders and metabolic syndrome [30], the value of visceral fat area associated with an increased risk of obesity-related disorders was 103.8 cm^2^.

A previous investigation with a sample of 233 middle-aged and older women (45 to 73 years) showed that a visceral fat area of ≥106 cm^2^ is associated with elevated risks for having low HDL cholesterol concentrations, hypertriglyceridemia, a high LDL/HDL cholesterol ratio, impaired glucose tolerance, and hyperinsulinemia [47], with a visceral fat area of ≥163 cm^2^ being predictive of even greater risks for metabolic risk factors for coronary heart disease when compared to lower visceral fat levels.

In relation to the values presented above, in our study, males had healthy visceral fat levels at both measure points (94.55–97.27), while females, despite having a significant reduction in their mean visceral fat area, continued to be at an elevated risk for cardiometabolic disorders (126.60–114.91).

### 4.5. Physiological Health Variables as Predictors of Attendance

Before discussing which physiological health variables predicted attendance, we would like to point out that in our study, females registered a larger increase in skeletal muscle mass and a larger decrease in body fat when compared to males. While age predicted a decrease in skeletal muscle mass in older individuals, a reduction in skeletal muscle mass was expected for older individuals since the association of aging and progressive muscle loss are well stablished [48], with skeletal muscle mass decreasing at a rate of approximately three to eight percent per decade after 30 years, and an even higher rate of decline after the age of 60 [49]. These results, albeit outside our aim, are important to understand further analyses.

Although the results showed differences in physiological health from T1 to T2 within individuals for the variables analyzed, the attendance percentages did not have any associations with changes in physiological variables. However, cardiorespiratory fitness had a significant relationship with an individual’s attendance, i.e., individuals with better cardiorespiratory fitness at baseline had higher attendance rates. Other studies have shown similar results, as seen in a previous review performed to determine exercise adherence rates and their predictors in 21 randomized control trials [50]. As per our results, individuals with better cardiorespiratory fitness at baseline had the best adherence to physical training.

Nevertheless, one of the challenges is to increase adherence to physical activity interventions. Some studies indicate that, among different populations, the more physically fit a person is at the start of the program, the higher attendance rate they have [50,51]. Since asylum seekers in general present lower fitness levels when compared to native populations, it could be speculated that the group may not have a high adherence rate to physical activity interventions.

### 4.6. Strengths and Limitations

The study’s strength was first and foremost its longitudinal study design with a predefined set of specific variables and objective measures. It allowed for an insight into physiological health developments over a 10-week period in a rarely studied population.

The primary methodological limitation of the study refers to the lack of randomization and a control group providing a standard for comparison when measuring the physiological outcomes. This limitation did not allow us to draw definitive conclusions on the effects of physical activity in the group studied; thereby, the results can only be interpreted as associations.

Moreover, the mixed models we applied w particularly useful in longitudinal studies and are often preferred to other approaches because they can be used with missing values. Nevertheless, even if in our models the covariates’ intercepts effects were fixed and individuals’ intercepts were set at random in order to test the differences within individuals with regard to the dependent variables, the lack of a control group still presents a limitation.

Upon first glance, our results might suggest that a short physical activity intervention combining resistance and aerobic training yields small but positive results in physiological variables for this specific population. For instance, a short intervention (8 to 10 sessions in a 10-week period, for example) would allow a much larger number of people to take part of the training program and still have positive impacts on their health. Indeed, the amount of training in “Health for Everyone” was dictated by financial and logistical constraints. However, the amount of physical activity that is recommended for adults is significantly higher. In order to improve and sustain cardiorespiratory fitness and reduce the risk of non-communicable diseases, the WHO [52] recommends at least 150 min of moderate-intensity physical activity per week or at least 75 min of vigorous-intensity physical activity per week. Therefore, the volume of exercise recommended is considerably higher than the volume that was provided by the intervention. Hence, even if a short intervention would be economically and logistically more feasible and still give positive effects, it would probably not have any long-term physiological effects.

In addition, socioeconomic variables, which could potentially have affected the participants’ physiological outcomes, were not taken into consideration in the present study. The participants’ educational levels were not analyzed, and as reported in previous studies [53,54], lower educational levels are linked to higher risks of physical inactivity and a higher incidence of non-communicable diseases [55].

Income levels and participation in the workforce were also not examined in this study. Higher physical activity levels are noted among those with higher income and a steady source of revenue. Meanwhile, less disposable income and unemployment are strongly related to low physical activity levels [56]. Even though information on income levels was not collected, it could be assumed that the vast majority of the participants were still navigating the Swedish immigration system and relied heavily on subsidies and financial support from the Swedish state, since the recruitment was carried out by Blekinge’s municipalities within the framework of social orientation for asylum seekers from countries currently experiencing armed conflicts.

Although this study provides a valuable insight into asylum seekers’ physiological health, due to the relatively small number of participants in the project, we do not aim to use the cardiorespiratory fitness values found in this investigation to set reference values for VO2 max for asylum seekers in Sweden. Based on its limitations and study design, caution should be applied when arriving to conclusions based on the study findings.

## 5. Conclusions

The results from our study are in accordance with previous research on the associations of physical activity interventions in the physical health of adult populations. The participants showed significant improvements in physiological health variables of physical fitness and body composition, most noticeably in a significant increase in cardiorespiratory fitness. Moreover, individuals with higher initial cardiorespiratory fitness levels were more likely to adhere to the intervention, which leads to the assumption that participants who were already physically active were more inclined to maintain a physically active lifestyle.

Given the current format and the limitations of the project “Health for Everyone-Sport, Culture and Integration”, it would be of interest to investigate whether an extended version could lead to long-term improvements in the participants’ physiological health and whether it could beneficially impact their exercise attitudes and their psychological and social health.

With the complexity of asylum seekers’ physical and mental health needs, barriers and facilitators could also be identified in order to increase participation in a physical activity intervention and achieve more significant and lasting results. For example, Haith-Cooper et al. [57] found that, among asylum seekers, stress, poverty, and temporary living conditions acted as barriers for participating in physical activity.

A point to be explored in future studies is the potential effects physical activity interventions may have on immigrant populations in the context of language acquisition. Evidence shows that physical activity interventions in young asylum seekers have a positive impact on their second language learning outcomes [58]. Since higher physical fitness is associated with improved cognition and literacy [59], it may therefore facilitate integration into a new society.

Overall, the results from this study provide an insight into asylum seekers’ health status and could serve as a base for implementing an intervention scale-up, where culturally sensitive approaches to physical activity are used to improve vulnerable populations’ physical and psychological health, and act as a guide for future policies towards health equality and inclusion in society.

## Figures and Tables

**Table 1 brainsci-12-00822-t001:** Descriptive characteristics of the study sample.

Variables	Gender	Mean/SD
Age	Female	40.29 ± 9.34
	Male	39.28 ± 10.16
	Total	39.70 ± 9.81
Attendance Percentage	Female	70% ± 0.33
	Male	76% ± 0.26
	Total	73% ± 0.29
Differences between females and males in attendance percentage	*t*-value	*p*-value
	1.99	0.05

Note: min: minimum, max: maximum, SD: standard deviation.

**Table 2 brainsci-12-00822-t002:** Physiological health measures at week 0 (T1) and week 10 (T2).

Physiological Health	Gender	T1 Mean (SD)	Total Mean T1 (SD)	T2 Mean (SD)	Total Mean T2 (SD)	Mean Change T1 to T2
Cardiorespiratory Fitness (VO_2_ max; mL/min/kg)	Female	28.58 (SD = 4.91)	31.46 (SD = 5.97)	31.22 (SD = 4.87)	34.35 (SD = 6.42)	2.64
	Male	33.54 (SD = 5.81)		36.44 (SD = 6.49)		2.90
Body Weight (kg)	Female	69.41 (SD = 11.93)	75.81 (SD = 14.64)	68.35 (SD = 10.29)	75.95 (SD = 13.98)	−1.06
	Male	80.37 (SD = 14.71)		80.83 (SD = 13.88)		0.46
BMI (kg/m^2^)	Female	26.57 (SD = 4.23)	26.56 (SD = 4.34)	26.34 (SD = 3.72)	26.63 (SD = 4.09)	−0.23
	Male	26.55 (SD = 4.43)		26.81 (SD = 4.31)		0.26
Skeletal Muscle Mass (%)	Female	34.60 (SD = 3.50)	39.04 (SD = 5.51)	34.98 (SD = 3.46)	39.54 (SD = 5.52)	0.38
	Male	42.21 (SD = 4.39)		42.45 (SD = 4.53)		0.24
Body Fat Mass (%)	Female	36.69 (SD = 6.38)	30.01 (SD = 9.07)	35.96 (SD = 6.15)	29.21 (SD = 8.90)	−0.73
	Male	25.26 (SD = 7.58)		24.91 (SD = 7.63)		−0.35
Visceral Fat (cm^2^)	Female	126.60 (SD = 44.10)	107.91 (SD = 45.57)	114.96 (SD = 39.42)	104.53 (SD = 43.50)	−11.64
	Male	94.55 (SD = 41.83)		97.27 (SD = 44.84)		2.72

Note: SD: standard deviation.

**Table 3 brainsci-12-00822-t003:** Linear mixed model analysis of the predictors of physiological health.

Variables	Model 1				Model 2 (Random Effects)				Model 3 (Fixed Effects)			
	Est.	SE	*p*-Value	ICC	AIC	Est.	SE	*p*-Value	AIC	Est.	SE	*p*-Value
Cardiorespiratory Fitness (VO_2_ max)	32.38	0.33	0.00	0.80	3115.81	50.93	8.91	0.00	3222.57	54.13	7.97	0.00
Gender						−9.01	5.52	0.10		−10.70	4.98	0.03
Age						−0.43	0.24	0.07		−0.50	.22	0.02
Attendance Percentage						5.39	11.24	0.63		3.45	9.85	0.727
BMI	26.61	0.22	0.00	0.98	2664.25	20.58	6.94	0.00	3455.94	22.96	5.75	0.00
Gender						−0.15	4.34	0.97		−1.84	3.70	0.62
Age						0.22	0.19	0.24		0.20	0.16	0.20
Attendance Percentage						2.20	8.81	0.80		−1.08	7.15	0.88
Body Fat Percentage	29.90	0.47	0.00	0.97	3492.50	−4.59	12.14	0.71	4057.93	−5.21	9.36	0.58
Gender						17.57	7.55	0.02		17.77	6.01	0.00
Age						0.56	0.30	0.06		0.65	0.26	0.01
Attendance Percentage						−2.45	15.39	0.87		2.62	11.65	0.82
Body Weight (kg)	75.97	0.75	0.00	0.99	4020.39	79.26	24.14	0.00	4847.40	88.60	11.12	0.00
Gender						−14.48	15.01	0.34		−20.20	11.65	0.08
Age						0.56	0.60	0.35		0.51	0.50	0.31
Attendance Percentage						0.66	30.63	0.98		−9.61	22.52	0.67
SMM Percentage	39.12	0.29	0.00	0.98	2830.52	61.28	6.90	0.00	3380.54	62.45	5.35	0.00
Gender						−12.08	4.29	0.01		−12.64	3.43	0.00
Age						−0.34	0.17	0.05		−0.41	0.15	0.00
Attendance Percentage						1.92	8.75	0.83		−1.99	6.66	0.77
Visceral Fat	107.66	2.34	0.00	0.98	5576.84	3.62	73.55	0.96	6331.85	67.66	55.91	0.23
Gender						30.16	45.76	0.51		10.62	35.45	0.77
Age						2.17	1.83	0.24		0.27	1.52	0.86
Attendance Percentage						−40.44	93.34	0.67		−58.05	70.71	0.41

Note: Est.: regression coefficient, SE: standard error, SMM: skeletal muscle mass. ML: maximum likelihood. ICC: intraclass correlation coefficients. AIC: Akaike Information Criteria.

**Table 4 brainsci-12-00822-t004:** Effect size, SEM, and MDC for all physiological health measures.

Physiological Health	Gender	Cohen’s *f^2^*	SEM	MDC_95_
Cardiorespiratory Fitness (VO_2_ max; mL/min/kg)	Female	0.04	1.38	1.96
	Male	0.04		
Body Weight (kg)	Female	0.01	1.46	1.69
	Male	0.01		
BMI (kg/m^2^)	Female	0.01	0.43	0.92
	Male	0.01		
Skeletal Muscle Mass (%)	Female	0.03	0.55	1.04
	Male	0.02		
Body Fat Mass (%)	Female	0.02	0.91	1.34
	Male	0.01		
Visceral Fat (cm^2^)	Female	0.02	4.56	2.99
	Male	0.02		

Note: SEM: standard error of measurement, MDC_95_: minimum detectable change at 95% confidence interval.

**Table 5 brainsci-12-00822-t005:** Linear regression analysis of the predictors of attendance percentage.

Parameter	*B*	Std. *β* Estimate	Std. Error	*p*-Value
Cardiorespiratory fitness (VO_2_ max)	0.008	0.171	0.003	0.003
BMI	0.000	0.004	0.004	0.947
Body Weight	5.639	0.003	0.001	0.957
SMM Percentage	0.004	0.082	0.003	0.123
Body Fat Percentage	−0.003	−0.079	0.002	0.140
Visceral Fat	0.000	−0.067	0.000	0.214
Age	0.002	0.071	0.002	0.171
Gender	−0.060	−0.102	0.030	0.048

## Data Availability

The data supporting the findings of this study are available from the research group, but restrictions apply to the availability of these data, and the data are not publicly available.

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
