# Peer review of "Effects of a 10-Week Physical Activity Intervention on Asylum Seekers’ Physiological Health"

_brainsci, 2022, doi:10.3390/brainsci12070822_

Round 1

Reviewer 1 Report

The authors sought to analyze two specific points: first, whether physical activity, practiced over several weeks, could have an impact on the physiological health of asylum seekers and second, whether the initial physiological status of individuals can predict their adherence to the training intervention. The project began in 2016 and ran for two years. A total of 467 individuals were enrolled, with equal numbers of women and men. Subjects performed a combination of resistance aerobic training for 8 sessions. Physiological data were recorded. Subjects completed demographic and psychological questionnaires at the beginning and end of the study.

The topic of the study is interesting because it addresses a public health issue for migrants; however, it does not seem to me to directly address the scope of the Brain Sciences journal. In addition, a number of key points suggest that the work needs to be significantly improved before to be re-evaluated for publication

I propose a few substantive comments to improve data analysis and discussion of the study:

The lack of a control group is a strong blocking point in my opinion. The authors are aware of this because they specify this point in the limitations section of the study. It is clear that it is often difficult in this type of longitudinal "field" analysis to find ‘control’ subjects with identical physiological and demographic characteristics. In the case of the present study, the absence of a control group is problematic because the differences reported by the authors in some physiological parameters are very small when comparing pre- and post-intervention data and in fact, it is difficult to interpret if the effects are actually related to the training or to confounding factors. For examples, at the beginning of the Discussion, authors proposed ‘The results confirmed the hypothesis that the “Health for everybody” intervention was associated with a beneficial impact on the asylum seekers’ physiological variables for both males and females, with improvements in their body composition and cardiorespiratory fitness.’ or (lines 278-280) authors noted ‘The participants had a significant increase in cardiorespiratory fitness from T1 (week 0) to T2 (week 10). At the end of the intervention, males had an increase in VO2 max of 2.96 ml/min/kg (M=36.55, SD=6.49) and females an increase of 2.57 ml/min/kg (M= 31.17, 280 SD=4.87)’. 

On what statistical basis do the authors base this statement? If you are only relying on the linear mixed model, why do you also propose Table 2? No statistical difference is shown in Table 2. No comment is made on this table 2 which is yet at the basis of the study. Moreover, the percentages of variation of the physiological markers are very low (at most 9% for VO2max, between 0.5 and 2% for the body composition markers). It’s difficult to consider that there could have a significant difference between pre- and post-training intervention! The authors' proposal to perform a mixed linear analysis is interesting, but again it would need to be performed also with a control group. Mixed models are particularly useful in situations where repeated measurements are made on the same variables (longitudinal study) as in this paper. They are often preferred to other approaches such as rANOVA, as they can be used with missing values. Nevertheless, very often mixed models consider, in addition to the fixed effects, random effects which allow to reflect the correlation between the statistical units. What about fixed versus random effects? This point could be further clarified.

Lines 381-381 authors noted a limitation: ‘Due to financial and logistical constrains, the participants underwent approximately one hour of physical training per week, which is to be considered suboptimal.

If results of the study confirm significant improved fitness outcomes for this population of asylum seekers, the fact that these subjects practiced "only" one hour per week cannot be considered as suboptimal. Indeed, the authors could argue that a short practice of combined resistance and aerobic training already allows to obtain suitable results for the health of this public. This is of great interest in terms of psycho-physiological health monitoring for these people in precarious situations. Moreover, it could be original to discuss about "minimal/short" intervention/training that allow triggering consecutive weak signals of this intervention (first effects even before the larger significant effects) on the subjects' condition.

Under these conditions (8 to 10 repetitions over a two-week period, for example), it would be possible to offer a much larger number of people a training program with positive impacts on their health. This last point could be important to discuss.

One question that remains to be answered is the memory of the effects of this type of training. Are 10 sessions enough to maintain an improved physiological state in the long term? This question should be discussed.

One question that remains to be answered is the memory of the effects of this type of training. Are 8-10 sessions sufficient to maintain an improved physiological state over the time? This question seems to me to be important in this study for this population and needs to be discussed.

General comments:

Some passages of the text are in bold. No reason for that I think!

Some typing errors are reported

The part "treatment of outliers and testing of the normality assumption" should appear first in the statistical section. Then after the statistical tests used to compare the pre- and post-intervention data values have to be detailed. This needs to be clarified, or done if it has not yet been done!

Author Response

Response to Reviewer 1 Comments

Point 1: I propose a few substantive comments to improve data analysis and discussion of the study:

The lack of a control group is a strong blocking point in my opinion. The authors are aware of this because they specify this point in the limitations section of the study. It is clear that it is often difficult in this type of longitudinal "field" analysis to find ‘control’ subjects with identical physiological and demographic characteristics. In the case of the present study, the absence of a control group is problematic because the differences reported by the authors in some physiological parameters are very small when comparing pre- and post-intervention data and in fact, it is difficult to interpret if the effects are actually related to the training or to confounding factors. For examples, at the beginning of the Discussion, authors proposed ‘The results confirmed the hypothesis that the “Health for everybody” intervention was associated with a beneficial impact on the asylum seekers’ physiological variables for both males and females, with improvements in their body composition and cardiorespiratory fitness.’ or (lines 278-280) authors noted ‘The participants had a significant increase in cardiorespiratory fitness from T1 (week 0) to T2 (week 10). At the end of the intervention, males had an increase in VO2 max of 2.96 ml/min/kg (M=36.55, SD=6.49) and females an increase of 2.57 ml/min/kg (M= 31.17, 280 SD=4.87)’. 

On what statistical basis do the authors base this statement? If you are only relying on the linear mixed model, why do you also propose Table 2? No statistical difference is shown in Table 2. No comment is made on this table 2 which is yet at the basis of the study. Moreover, the percentages of variation of the physiological markers are very low (at most 9% for VO2max, between 0.5 and 2% for the body composition markers). It’s difficult to consider that there could have a significant difference between pre- and post-training intervention! The authors' proposal to perform a mixed linear analysis is interesting, but again it would need to be performed also with a control group. Mixed models are particularly useful in situations where repeated measurements are made on the same variables (longitudinal study) as in this paper. They are often preferred to other approaches such as rANOVA, as they can be used with missing values. Nevertheless, very often mixed models consider, in addition to the fixed effects, random effects which allow to reflect the correlation between the statistical units. What about fixed versus random effects? This point could be further clarified.

Response to point 1:

Thank you for your comment. We agree in the fact that we have not specified Table 2 as clear as possible. We have added the following text to the Results (lines 217-221):

“Table 1 indicates the descriptive characteristics of the sample used in this study. Comparing mean differences in attendance percentage among females and males indicated that box genders participated in the physical activity sessions to roughly the same extent. Table 2 indicates the physiological health variables related to before (T1) and after (T2) the intervention.”

We have also clarified the basis of the statement regarding the results (306-309):

“The results of the linear mixed model confirmed that the “Health for everybody” intervention was associated with a beneficial impact on the asylum seekers’ physiological variables for both males and females, with improvements in their body composition and cardiorespiratory fitness.”

Regarding the question about the limitations of our proposed analyses without a control group, we took the liberty to use the Reviewer’s own words to point this out in the Strenghts and limitations section, we hope this is fine with the Reviewer:

“Moreover, the mixed models we applied are particularly useful in longitudinal studies and are often preferred to other approaches because they can be used with missing values. Nevertheless, even if in our models, the covariates’ intercepts effects were fixed, and individuals’ intercepts were set at random in order to test the differences within individuals with regard to the dependent variables, the lack of a control group still presents a limitation.”

In addition, fixed effects model and the Akaike Information Criteria (AIC) for both random and fixed effectes were aded to Table 3.

Point 2: Lines 381-381 authors noted a limitation: ‘Due to financial and logistical constrains, the participants underwent approximately one hour of physical training per week, which is to be considered suboptimal.

If results of the study confirm significant improved fitness outcomes for this population of asylum seekers, the fact that these subjects practiced "only" one hour per week cannot be considered as suboptimal. Indeed, the authors could argue that a short practice of combined resistance and aerobic training already allows to obtain suitable results for the health of this public. This is of great interest in terms of psycho-physiological health monitoring for these people in precarious situations. Moreover, it could be original to discuss about "minimal/short" intervention/training that allow triggering consecutive weak signals of this intervention (first effects even before the larger significant effects) on the subjects' condition.

Under these conditions (8 to 10 repetitions over a two-week period, for example), it would be possible to offer a much larger number of people a training program with positive impacts on their health. This last point could be important to discuss.

One question that remains to be answered is the memory of the effects of this type of training. Are 10 sessions enough to maintain an improved physiological state in the long term? This question should be discussed.

Response to point 2: We agree with the reviewer statement that a short practice of combined resistance and aerobic training allows for positive changes in the participants’ physiological variables. However, we conclude that a more significant change in their physiological health would occur with a prolongued version of the physical activity intervention (as stated in section 5. Conclusions). Thant being said, we found the Reviewer’s points relevant and have addes some of them to the manuscript (lines 393-406):

“At first sight, our results might suggest that a short physical activity intervention combining resistance and aerobic training yields small but positive results in physio-logical markers for this specific population. For instance, a short intervention (8 to 10 sessions in a 10-week period, for example) would allow a much larger number of people to take part of the training program and still have positive impacts on their health. Indeed, the amount of training in “Health for Everyone” was dictated by financial and logistical constrains. However, the amount of physical activity that is recommended for adults is significantly higher. In order to improve and sustain cardiorespiratory fitness and reduce the risk of noncommunicable diseases, WHO [48] recommends at least 150 minutes of moderate-intensity physical activity per week or at least 75 minutes of vigorous-intensity physical activity per week. Therefore, the volume of exercise recommended is consid-erably higher than the volume provided by the intervention. Hence, even if a short intervention would be economically and logistically more feasible and still give positive effects, it would probably not have any long-term physiological effects.” 

Point 3: General comments

Some passages of the text are in bold. No reason for that I think!

Some typing errors are reported.

The part "treatment of outliers and testing of the normality assumption" should appear first in the statistical section. Then after the statistical tests used to compare the pre- and post-intervention data values have to be detailed. This needs to be clarified, or done if it has not yet been done!

Response to point 3: Bold text passages were corrected, along with typing errors. We have also moved the outlier text as suggested. Regarding data to compare pre- and post-intervention values, total means are now displayed in Table 2.  

Reviewer 2 Report

Thank you very much for the opportunity to review this manuscript.

I believe that the topic addressed in the manuscript is of great interest and topicality, it is true that there are not many physical activity interventions in people seeking asylum, however, the manuscript has some areas for improvement that I detail below.

Introduction

The introduction I think follows an adequate thread, however, I believe that it does not include important and current manuscripts for the subject such as:

Haith-Cooper, M., Waskett, C., Montague, J. et al. Exercise and physical activity in asylum seekers in Northern England; using the theoretical domains framework to identify barriers and facilitators. BMC Public Health 18, 762 (2018). https://doi.org/10.1186/s12889-018-5692-2.

Michelini, E. (2020). Refugees, physical activity and sport: a systematic literature review. Refugees, physical activity and sport: a systematic literature review, 131-152.

Nilsson, H., Gustavsson, C., Gottvall, M., & Saboonchi, F. (2021). Physical activity, post-traumatic stress disorder, and exposure to torture among asylum seekers in Sweden: a cross-sectional study. BMC psychiatry, 21(1), 1-12.

Michelini, E. (2022). Organised sport in refugee sites: An ethnographic research in Niamey. European Journal for Sport and Society, 19(1), 1-17.

Materials and methods.

The study design is not described, it is necessary to include it. The absence of a control group, in my opinion, is the major weakness of the study, although it is indicated in the limitations of the manuscript, it is a major limitation.

The sample calculation or, failing that, the statistical power obtained with the sample included in the study is not included.

The minimum real change in the measurements included in the manuscript is not included; it is necessary to include it in order to be able to interpret the data adequately; a change can only be considered to have occurred if it exceeds the minimum real change.

Results

The effect size for each of the measures is not included.

Discussion

It is necessary to include in the discussion some important quotations on the subject (see the recommended quotations in the introduction).

Conclusion

In order to know if the variables studied have really improved, it is important to know if they have exceeded the minimum real change; without this information it cannot be affirmed that there is a change. 

Author Response

Response to Reviewer 2 Comments

Point 1: Introduction

The introduction I think follows an adequate thread, however, I believe that it does not include important and current manuscripts for the subject such as:

Haith-Cooper, M., Waskett, C., Montague, J. et al. Exercise and physical activity in asylum seekers in Northern England; using the theoretical domains framework to identify barriers and facilitators. BMC Public Health 18, 762 (2018). https://doi.org/10.1186/s12889-018-5692-2.

Michelini, E. (2020). Refugees, physical activity and sport: a systematic literature review. Refugees, physical activity and sport: a systematic literature review, 131-152.

Nilsson, H., Gustavsson, C., Gottvall, M., & Saboonchi, F. (2021). Physical activity, post-traumatic stress disorder, and exposure to torture among asylum seekers in Sweden: a cross-sectional study. BMC psychiatry, 21(1), 1-12.

Michelini, E. (2022). Organised sport in refugee sites: An ethnographic research in Niamey. European Journal for Sport and Society, 19(1), 1-17.

Response to point 1:

The article by Haith-Cooper et al. (2018) was included in the section 5. Conclusions

The article by Michelini (2020) contains 26 studies selected for a systematic review, with 15 papers belonging to the category of “health sciences group”, which is concerned with the physical, mental and social health of refugees. Since the article analyses qualitative data gathered mostly through focus group discussions, interviews and field notes, the authors deemed other reference sources regarding the quantitative analysis of a predefined set of physiological variables more relevant.

The article by Nilsson et al. (2021) was reviewed for reference during the writing phase of the present manuscript. Since the article aims to determine the association between levels of physical activity and post-traumatic stress disorder (PTSD) symptoms severity, this particular reference will be used for a subsequent manuscript currently under work, which aims to analyse the participants’ psychological health variables before and after the 10-week intervention.

The article by Michelini (2022) refers ot organized sport in a refugee context and its application as a tool for social development, examing which roles organized sport has in refugee sites. Although organized sport utilizes physical activity during its practice; physical acitivty in a broader context does not imply the use of organized sport. The asylum seekers in our study did not participate in organized sport nor were placed in a refugee site, instead they engaged in circuit training physical activity programme, and were in the process of adjusting to their host country. Although the article provides a theoretical framework of how the dynamics of organized sport play in a refugee context, the authors deemed other literary references as more relevent in the context of the present manuscript.

Point 2: Materials and methods.

The study design is not described, it is necessary to include it. The absence of a control group, in my opinion, is the major weakness of the study, although it is indicated in the limitations of the manuscript, it is a major limitation.

The sample calculation or, failing that, the statistical power obtained with the sample included in the study is not included.

The minimum real change in the measurements included in the manuscript is not included; it is necessary to include it in order to be able to interpret the data adequately; a change can only be considered to have occurred if it exceeds the minimum real change.

Response to point 2: the study design is included in the text in section 2.3. Statistical analysis:

“The study had an interventional and longitudinal design. In short, set of physiological variables was measured before and after the 10-week physical activity intervention for each participant”. Statistical power was added in page 5, line 214. Minimum detectable change (MDC) was added, please see pages 7 and 8.

Point 3: Results

The effect size for each of the measures is not included.

Response to point 3: effects size was added in section “3.3 Effect size and minimum detectable change calculation for each physiological measure”, page 7.

Point 4: Discussion

It is necessary to include in the discussion some important quotations on the subject (see the recommended quotations in the introduction).

Response to point 4: Thank you for this. We added Haith-Cooper et al. (2018).

Point 5: Conclusion

In order to know if the variables studied have really improved, it is important to know if they have exceeded the minimum real change; without this information it cannot be affirmed that there is a change. 

Response to point 5: regarding the reviewer’s comments on minimum change, section 3.3 Effect size and minimum detectable change calculation for each physiological measure, was added to the manuscript, which addresses minimum detectable change (MDC) and interpretation. The text was added to pages 7 and 8, including but not limited to the excerpt: “Changes in physiological health measures were also estimated through Minimum detectable change (MDC)—a statistical estimate of the smallest amount of change that can be detected by a measure that corresponds to a noticeable change in the variable under study over time and not related to measurement error”, with outputs being added to table 4.

Round 2

Reviewer 1 Report

Thank you for the corrections and answers to my questions. The manuscript now seems quite acceptable for publication.

Reviewer 2 Report

The authors have responded to all my comments. I believe that the manuscript can be accepted in its present form.

This manuscript is a resubmission of an earlier submission. The following is a list of the peer review reports and author responses from that submission.

Round 1

Reviewer 1 Report

I have read your article with interest. The aim of the article is to evaluate the influence health project on the asylum seekers and if physiological health variables at baseline can predict adherence to the intervention. I have made some comments below.

  • Choose the numbers either the two-digit or the one-digit approximation (e.g., line 105), not both, and keep it constant in the Entire document.
  • The project “Health for Everyone - Sport, Culture and Integration” was introduced in the Introduction part. However, as the only health intervention in this study, the detailed information of the intervention was basically not mentioned. For example, what's the content of the intervention, and how it was carried out, etc. It is suggested that this information should be provided.
  • This intervention was carried out in 10 weeks, it is suggested that the reason why this study choose 10 weeks should be made clear. Is it because the project only last for 10 weeks, or based on relevant literature?
  • Please describe the criteria for exclusion of study subjects.

Author Response

1. Choose the numbers either the two-digit or the one-digit approximation (e.g., line 105), not both, and keep it constant in the entire document.

Matheus Guerra: The number approximations regarding the statistical analysis' results presented in this manuscript was corrected according to the reviewer's request.

2. The project “Health for Everyone - Sport, Culture and Integration” was introduced in the Introduction part. However, as the only health intervention in this study, the detailed information of the intervention was basically not mentioned. For example, what's the content of the intervention, and how it was carried out, etc. It is suggested that this information should be provided.

Matheus Guerra: A detailed description of the intervention's contents and activities is provided in section 2.Materials and methods.

3. This intervention was carried out in 10-weeks, it is suggested that the reason why this study choose 10 weeks should be made clear. Is it because the project only last for 10 weeks, or based on relevant literature?

Matheus Guerra: Revision made in lines 106-107.

4. Please describe the criteria for exclusion of study subjects.

Matheus Guerra: The study included only individuals who are asylum seekers from countries currently experiencing armed conflicts.

Reviewer 2 Report

Dear authors, congratulations on your interesting work, aiming to improve the quality of life of asylum seekers. Science should induce real change in the life of those in need. 

I have a few suggestions for you:

Throughout the document, please consider using "native" instead of "indigenous populations". Line 92, please describe the associations you would like to investigate. Line 123, please give details on the "other self-reports of validated psychological measures that was used. Line 129/130, please rewrite the sentence, it is not clear. Line 152, the height of the participants was self-reported or measured? I did not find information on that.

Line 204/205, I do believe that removing 93 extreme outliers induce a major bias in the work. Personally, I don't agree with this level of pruning of the data. I believe you will not change this now, but I suggest you remove this from the document.

Author Response

1. Throughout the document, please consider using "native" instead of "indigenous populations".

Matheus Guerra: Modifications in the text were done as per reviewer's request.

2. Line 92, please describe the associations you would like to investigate.

Matheus Guerra: Modifications in the text were done as per reviewer's request.

3. Line 123, please give details on the "other self-reports of validated psychological measures that was used.

Matheus Guerra: Modifications in the text were done as per reviewer's request.

4. Line 129/130, please rewrite the sentence, it is not clear.

Matheus Guerra: Modifications in the text were done as per reviewer's request

5. Line 152, the height of the participants was self-reported or measured? I did not find information on that.

Matheus Guerra: Modifications in the text were done as per reviewer's request

6. Line 204/205, I do believe that removing 93 extreme outliers induce a major bias in the work. Personally, I don't agree with this level of pruning of the data. I believe you will not change this now, but I suggest you remove this from the document.

Matheus Guerra: we understand your comment and prefer to keep this information in the text. This manuscript will be the first of a series of studies and your comments will be taken into consideration in our further work.

Reviewer 3 Report

ijerph-1700389_review

Title: Effects of a 10-week Physical Activity Intervention on Asylum Seekers’ Physiological Health

Comments and Suggestions for authors

Dear authors,

I have carefully read your paper, which studies the associations of a ten-week physical activity intervention with the physiological health of asylum seekers and investigates whether variables in physiological health at baseline could predict adherence to the intervention. The authors concluded that participants showed significant improvements in physiological health variables. Moreover, individuals with higher initial cardiorespiratory fitness levels were more likely to adhere to the intervention.

In general, the manuscript is well-written. The text is understandable and organized. In my opinion, the introduction, results and discussion sections are well-described and analyses are appropriate However, I found several issues in materials and methods section and in the research design that should be addressed to improve the paper, in my opinion.

Specific comments:

Introduction

Introduction section is adequate and complete.

Material and methods

I found several issues in materials and methods section and in the research design that should be addressed.

- Page 2-3. You mentioned in lines 87-88 in the introduction section that “Within the project, asylum seekers were offered physical activity once a week during a 10-week period in groups of 20 to 30 individuals”.

How were the groups configured? Were they mixed groups of men and women? Or did the men train in a specific way and the women in another? Did all groups follow the same exercise protocol?

How did you control during the time that the intervention lasted that the participants did not perform any type of training or additional exercise? This is key and fundamental to your study, as this could be influencing your results.

-  Page 2-3, lines 99-105. Please, could you add information about the period of time in which the data were recorded, places where the training and recovery program were carried out (i.e.: university, hospital…?)

- Page 3, line 102-105: What inclusion or exclusion criteria were established to select the participants in the study?

Page 3 line 106-109: This information is central to your manuscript. It is necessary that you add detailed information about the intervention carried out. Please also add references that support this information and the selected intervention.

- Page 3, line 115. Ethical considerations for the study were not mentioned, only that informed consent was obtained, was there institutional ethical approval for the study?

Page 3, line 124-126: You carried out an intervention that consists of 8 exercise sessions, I understand that the first and last session are dedicated to taking measurements, therefore, in reality, your intervention is 8 weeks and not ten, please clarify this. Keep this in mind for the title and other mentions in this manuscript.

- Pages 4, lines 152-153: Report the reliability and validity the direct segmental multi-frequency bioelectrical impedance analysis (DSM-BIA) an InBody 720 153 body composition analyzer. Please add some supporting reference, if possible.

- Page 4, line 178. There is no statement about the sample size. How were the sample size calculated? This information is very important to support the external validity of the study findings

- Page 4, line 170: I suggest you that include the acronym of “BS test” the first time the full term "back squat test" appears in the text, specifically on page 3 line 105. Please check it.

-Page 4, line 178: I have observed that reference number 52 of the list of references is not cited in the text. I think this reference could support this paragraph. Please check it. If necessary, add another reference.

- Page 5, lines 199-207. Please report the reliability and validity for the 5-RM test if possible, add some reference that supports it, if necessary.

- Page 5, line 221. You mention in the results section that in the present study the performance of the EE was ≈93% of the recommended volume. How was the recommended volume determined? Please, add this information, is possible.

Results

The results section is well-structured and comprehensive. I suggest performing a flow diagram to facilitate monitoring the entire study participants.

  • Page 5, lines 214-217: I suggest adding more information about descriptive data of the participants, for example: country of origin, if they previously practiced some exercise, habits related to diet, tobacco and alcohol…..
  • Tables: I suggest that you add the n for each group in all the tables.

Discussion

Your discussion section is in general adequate and complete.

Conclusions

I think conclusions should be rephrased in a more careful way. The conclusions must respond to the objectives of your study. You said that “Study participants showed significant improvements in physiological health variables of physical fitness and body composition, most noticeably by a significant …..lifestyle”. As you recognised, based on the limitations and design of your study, caution should be applied when arriving to conclusions based on the study findings. Please clarify it, i.e: The results of our study seem to….

I hope that my comments could help to improve the paper. Congratulations for your research.

Author Response

1. Page 2-3. You mentioned in lines 87-88 in the introduction section that “Within the project, asylum seekers were offered physical activity once a week during a 10-week period in groups of 20 to 30 individuals”. How were the groups configured? Were they mixed groups of men and women? Or did the men train in a specific way and the women in another? Did all groups follow the same exercise protocol?

Matheus Guerra: Modifications in the text were done as per reviewer's request.

2. How did you control during the time that the intervention lasted that the participants did not perform any type of training or additional exercise? This is key and fundamental to your study, as this could be influencing your results.

Matheus Guerra: We did not control for additional training and/or exercise.

3. Page 2-3, lines 99-105. Please, could you add information about the period of time in which the data were recorded, places where the training and recovery program were carried out (i.e.: university, hospital…?)

Matheus Guerra: this information is detailed in the section 2.1. Participants and procedure, lines 100-112.

4. Page 3, line 102-105: What inclusion or exclusion criteria were established to select the participants in the study?

Matheus Guerra: The recruitment was carried by Blekinge’s municipalities, being scheduled within the framework of the social orientation for asylum seekers from countries currently experiencing armed conflicts, lines 103-105.

5. Page 3 line 106-109: This information is central to your manuscript. It is necessary that you add detailed information about the intervention carried out. Please also add references that support this information and the selected intervention.

Matheus Guerra: A detailed description of the intervention, measures, testing and references are provided in sections 2.1 Participants and procedure and 3. Methods.

6. Page 3, line 115. Ethical considerations for the study were not mentioned, only that informed consent was obtained, was there institutional ethical approval for the study?

Matheus Guerra: Ethical approval is mentioned in lines 463-466.

7. Page 3, line 124-126: You carried out an intervention that consists of 8 exercise sessions, I understand that the first and last session are dedicated to taking measurements, therefore, in reality, your intervention is 8 weeks and not ten, please clarify this. Keep this in mind for the title and other mentions in this manuscript.

Matheus Guerra: The authors respectfully disagree with this statement. The total intervention is 10 weeks, which counts for the period of time when the participants dedicated their time and effort to join the programme and the activities offered. The intervention consisted of different activities, and comprised testing at baseline and end-point, along with 8 training sessions in a total of 10 weeks.

8. Pages 4, lines 152-153: Report the reliability and validity the direct segmental multi-frequency bioelectrical impedance analysis (DSM-BIA) an InBody 720 153 body composition analyzer. Please add some supporting reference, if possible.

Matheus Guerra. Supporting references are in lines 159-160.

9. Page 4, line 178. There is no statement about the sample size. How were the sample size calculated? This information is very important to support the external validity of the study findings.

Matheus Guerra: In total, 467 individuals (263 males and 204 females) were enrolled in the project (lines 104-105).

10. Page 4, line 170: I suggest you that include the acronym of “BS test” the first time the full term "back squat test" appears in the text, specifically on page 3 line 105. Please check it.

Matheus Guerra: This comment may refer to another study and it is not applicable to this manuscript.

11. Page 4, line 178: I have observed that reference number 52 of the list of references is not cited in the text. I think this reference could support this paragraph. Please check it. If necessary, add another reference.

Matheus Guerra: line 178 regards the statistical analysis while reference number 52, which is mentioned in the text (line 400), regards to lower disposable income and unemployment being strongly related to low physical activity levels. We failed to see how this reference could benefit the above mentioned paragraph.

12. Page 5, lines 199-207. Please report the reliability and validity for the 5-RM test if possible, add some reference that supports it, if necessary.

Matheus Guerra: This comment may refer to another study and it is not applicable to this manuscript.

13. Page 5, line 221. You mention in the results section that in the present study the performance of the EE was ≈93% of the recommended volume. How was the recommended volume determined? Please, add this information, is possible.

 Matheus Guerra: This comment may refer to another study and it is not applicable to this manuscript.

14. The results section is well-structured and comprehensive. I suggest performing a flow diagram to facilitate monitoring the entire study participants.

Matheus Guerra: This suggestion will be taken into consideration in following studies.

15. Page 5, lines 214-217: I suggest adding more information about descriptive data of the participants, for example: country of origin, if they previously practiced some exercise, habits related to diet, tobacco and alcohol.

Matheus Guerra: This information was not recorded for this study.

16. Tables: I suggest that you add the n for each group in all the tables.

Matheus Guerra: This suggestion will be taken into consideration in following studies.